# Synthesis and Biological Activity of Glycosyl Thiazolyl Disulfides Based on Thiacarpine, an Analogue of the Cytotoxic Alkaloid Polycarpine from the Ascidian *Polycarpa aurata*

**DOI:** 10.3390/md23030117

**Published:** 2025-03-09

**Authors:** Dmitry N. Pelageev, Yuri E. Sabutski, Svetlana M. Kovach, Nadezhda N. Balaneva, Ekaterina S. Menchinskaya, Ekaterina A. Chingizova, Anna L. Burylova, Victor Ph. Anufriev

**Affiliations:** 1G. B. Elyakov Pacific Institute of Bioorganic Chemistry, Russian Academy of Sciences, Prospect 100 let Vladivostoku 159, Vladivostok 690022, Russia; alixar2006@yandex.ru (Y.E.S.); balaneva@piboc.dvo.ru (N.N.B.); ekaterinamenchinskaya@gmail.com (E.S.M.); martyyas@mail.ru (E.A.C.); anaburylova1@gmail.com (A.L.B.); anufriev@piboc.dvo.ru (V.P.A.); 2Institute of High Technologies and Advanced Materials, Far Eastern Federal University, Ajax Bay 10, Russky Island, Vladivostok 690922, Russia; svetakovach596403@gmail.com; 3Institute of the World Ocean, Far Eastern Federal University, Ajax Bay 10, Russky Island, Vladivostok 690922, Russia

**Keywords:** polycarpine, thiacarpine, bis(2-amino-4-phenyl-5-thiazolyl)disulfide, thioglycosides, *Polycarpa aurata*, cytotoxic activity, antimicrobial activity, biofilm formation, hemolysis

## Abstract

Polycarpine, a diimidazolyl disulfan alkaloid isolated from the ascidian *Polycarpa aurata*, showed high cytotoxic activity in vitro. However, in vivo experiments have shown that polycarpine has a high acute toxicity. At the same time, its synthetic thiazolyl analog, thiacarpine, showed less acute toxicity and had a greater therapeutic index, which makes its derivatives promising for further drug development. We assume that due to the presence of a disulfide bond in the molecules of polycarpine and thiacarpine and the possibility of its reduction in a living cell, the mercapto derivatives formed are responsible for the high activity of the original compounds. Based on this assumption, and to increase the selectivity of action, glycosyl disulfide conjugates of thiacarpine derivatives with thioglucose and thioxylose were synthesized and screened for their cytotoxic and antimicrobial activities. The target compounds did not show hemolytic activity at concentrations of up to 25 μM. Some of them exhibited moderate cytotoxic activity, blocked colony growth and migration of HeLa tumor cells, high antimicrobial activity, and inhibited biofilm formation comparable to or higher than that of a standard antibiotic (gentamicin) and antimycotic (nitrofungin).

## 1. Introduction

Polycarpine, a diimidazolyl disulfane alkaloid isolated from the ascidia of the genus *Polycarpa*, is the first representative of a new structural type of alkaloids containing 2-aminoimidazole rings linked by a disulfide bridge (Figure 1) [1,2,3,4]. In vitro experiments showed that polycarpine exhibited high cytotoxic and antitumor activity. However, in vivo experiments showed that it had acute toxicity. At the same time, its synthetic thiazolyl analog, thiacarpine, also exhibited a marked antitumor effect on both solid and ascites tumors. When injected intraperitoneally, it effectively inhibited the development of solid tumors such as LLC and melanoma B-16. However, unlike polycarpine, which was usually effective at high therapeutic doses, thiacarpine had a higher therapeutic index [5]. This makes its derivatives promising for further drug development. Moreover, having a wide pharmacological spectrum, 2-aminothiazole derivatives are important substances in medicinal chemistry and drug development [6,7]. Thus, 2-aminothiazole derivatives exhibit antitumor [8,9,10], antiviral [11], antimicrobial [12,13,14], anticonvulsant [15], antidiabetic [16], antiprion [17], and other properties [18]. For example, in vitro studies on the antitumor activity of various 2-aminothiazole derivatives have demonstrated their potent and selective inhibitory activity against a wide range of human cancer cell lines, such as breast, leukemia, lung, colon, central nervous system (CNS), melanoma, ovarian, kidney, and prostate [19,20,21].

The combination of different fragments in one hybrid molecule leads to a significant change in biological properties. Thus, the introduction of nonpolar fragments increases lipophilicity and, as a rule, increases cytotoxicity by facilitating diffusion through cell membranes. The introduction of polar fragments can increase the water solubility and bioavailability. One promising means of conjugation is the formation of disulfide bonds. They are of great importance in biological processes. Disulfide bonds are involved in the formation of the tertiary structure of proteins as well as in thiol-disulfide exchange with the participation of glutathione. Disulfides have also been reported to exhibit various biological activities. For example, some unsymmetrical aryl-alkyl disulfides inhibited the growth of methicillin-resistant *Staphylococcus aureus* and *Bacillus anthracis* [22]. Yoon and co-workers reported that some unsymmetrical disulfide compounds were inhibitors of *Mycobacterium tuberculosis* and *Haemophilus influenzae*, by interfering with acetohydroxyacid synthase (AHAS), a key enzyme in the biosynthesis pathway of branched-chain amino acids [23,24].

Unsymmetrical monoterpenyl-hetaryl disulfides based on heterocyclic disulfides and monoterpene thiols had good indicators of antioxidant, antimicrobial, and antiviral activities [25]. Some aromatic disulfides were reported to exhibit antiviral activity [26,27,28,29]. Moreover, glycosyl disulfides have attracted significant attention as glycomimetics with wide biological applications [30]. For example, they can be used as prodrugs that are activated in living cells by reducing disulfide bonds [31].

Glycoconjugation, that is, the binding of a drug with different sugars, is a promising approach that may increase the cellular uptake of the therapeutic compound. So, the idea of conjugating anticancer drugs to glucose residue is based on the Warburg effect. This effect describes the ability of cancer tissues to consume larger amounts of glucose than normal tissues. Overexpression of glycolytic enzymes and glucose transporters GLUTs mediates this effect and has been recognized as a hallmark of cancer development. The use of the Warburg effect as an anticancer strategy has attracted much attention, and the idea of using sugar-conjugated anticancer drugs to enhance the selectivity of their uptake has developed significantly over the past decade. Several compounds using the Warburg effect are undergoing advanced clinical trials, while many others are in preclinical development [32,33,34].

A similar effect on microorganisms can be expected for xylose. Many bacteria, including *E. coli*, tend to use active transport for D-xylose. In yeasts and fungi, transportation can occur by facilitated diffusion or active transport, and is highly dependent on species and whether it can be utilized to sustain growth. Yeast, such as *Pichia*, *Candida*, and *Debaryomyces*, have been reported to have xylose-proton symporters that are substrate-specific, have a high affinity, and are reliant upon induction by xylose for some species [35].

We assume that because of the presence of the disulfide bond in polycarpine and thiacarpine molecules and the possibility of its reduction in living cells, the mercapto derivatives formed as a result of this reduction are responsible for the high activity of the parent compounds. Based on this assumption, in order to study antimicrobial and cytotoxic activity in this work, we obtained thiazole-glucose and thiazole-xylose conjugates based on thiacarpine and its derivatives, in which the thiazole and carbohydrate fragments are connected through two sulfur atoms. The target compounds are expected to have high water solubility and bioavailability, and may also exhibit selectivity on tumor cells or pathogenic microorganisms.

## 2. Results and Discussion

### 2.1. Synthesis of Glycosyl Disulfides

Despite the development of numerous methods for the synthesis of asymmetric disulfides [30], it is challenging to find the most effective method for the substrates under consideration. Bis(2-amino-4-phenyl-5-thiazolyl)disulfides **1a–e**, thiacarpine and its analogues, were the starting compounds in our work because of the instability of mercapto derivatives **2a–e** during purification. These mercapto derivatives **2a–e** can be obtained by the reduction with sodium dithionite but they are readily oxidized back to the starting disulfides **1a–e** in air. Oxidation of a mixture of mercapto derivatives **2a–e** and **3a,b** with iodine gave the target acetylated asymmetric disulfides **4a–e** in satisfactory yields (Figure 1) [36]. The chosen method was found to be easy to perform and, considering the recovery of the starting disulfide (approximately 40%), the yields could be considered good.

The experiments with bis(2-amino-4-phenyl-5-thiazolyl)disulfide **1b** to test the typically effective alternative methods described in [37,38] showed that the use of sulfenyl halides **2b**’ and **3b**’ was ineffective (yield of about 15–19% for paths B and C compared to 28% for path A) and more difficult to perform, as required anhydrous solvents and had lower solubility of the starting disulfide (Figure 2).

Deacetylation of **4a–j** derivatives under the action of HCl in methanol gave glycosyl disulfides **5a–j** as hydrochlorides in excellent yields (Figure 1).

### 2.2. Biological Activity of the Compounds

#### 2.2.1. Cytotoxic Activity

The cytotoxic activities of the synthesized compounds were studied in tumor and normal human cells: HeLa (cervical cancer), Hep G2 (hepatocellular carcinoma), MDA-MB-231 (triple-negative breast cancer), MCF-7 (adenocarcinoma of the mammary gland ducts), PC-3 (prostate cancer), and BEAS-2B (bronchial epithelium). All compounds were also tested for their ability to destroy erythrocyte membranes. The experiments revealed that the studied compounds did not have hemolytic activity in the concentration range of up to 25 μM. The most stable cell lines were PC-3 and MCF-7, as none of the substances showed cytotoxic activity up to 25 μM during incubation for 24 h. As expected, acetylated derivatives **4a–j**, which are less polar substances, generally had higher toxicity than deacetylated derivatives **5a–j** and starting disulfides **1a–e**. Deacetylation mainly resulted in a marked decrease in cytotoxicity. The highest activity was observed for compounds with –OMe, –Me, and–Br substituents (Table 1). There was no pronounced effect of sugar backbone type on the cytotoxicity of the conjugates. It is worth noting that compound **4e** bearing an acetylated glucose moiety and a methyl substituent in the aromatic moiety showed the greatest cytotoxic effect on cancer HeLa cells (EC_50_ = 18.53 μM) and TNBC cells of the MDA-MB-231 line (EC_50_ = 14.95 μM). In addition, conjugate **4e** showed no cytotoxicity against normal cell culture BEAS-2B (up to 25 µM). Compounds **4c**, **4f**, and **4h** exhibited similar cytotoxic activity against HeLa and MDA-MB-231 tumor cells and were selected for further investigation.

#### 2.2.2. HeLa Cell Migration

Cell migration is an important process in cancer metastasis; therefore, the search for drugs that will help prevent this process remains relevant. The effects of these compounds on cell migration were evaluated using an in vitro wound scratch migration assay. In the control group, HeLa cells migrated to completely close the wound area after 24 h (Figure 2). However, in the treatment groups, cell migration into the wound areas was significantly reduced at 24 h in a dose-dependent manner. All selected compounds **4c**, **4e**, **4f**, **4h** showed similar migration inhibition effects. The maximum migration inhibition effect was observed for compounds **4e** and **4h** at a concentration of 10 µM, and was approximately 43% for both.

#### 2.2.3. Compounds Reduced the Number of Colonies Forming of HeLa Cells

To observe the long-term effects of the test compounds on HeLa cells, a colony formation assay was performed. The cells were treated with concentrations below the EC_50_ for 14 days. Colony formation was significantly reduced by all test compounds at the studied concentrations (Figure 3). The maximum inhibitory effect was observed at 10 µM, amounting to 76% for **4c** and 78% for **4f**. Compound **4f**, at a concentration of 5 µM, had less of an effect on colony formation. Thus, colony growth inhibition was approximately 16% of that of the control.

#### 2.2.4. Antimicrobial Activity

The antimicrobial activity of the synthesized series of disulfides was studied against the Gram-positive bacteria *Staphylococcus aureus*, Gram-negative bacteria *Escherichia coli*, and yeast-like fungi *Candida albicans* using a double dilution method in liquid nutrient media. The antimicrobial activity was assessed by changing the optical density of the medium using a plate spectrophotometer. There was no general pattern in the change in activity between the acetylated and deacetylated derivatives. This can be explained by the interaction of these hybrid structures with different targets in the living cells. These interactions can cause effects with different contributions and directions (increase or decrease toxicity). As a rule, deacetylation led to a decrease in activity; however, in the case of halogen derivatives **4c**, **4d**, and **4h**, the activity of the deacetylated products **5c**, **5d**, and **5h** was higher. Bromine (**4c**, **4h**, **5c**, and **5h**) and chlorine (**4d**, **4i**, **5d**, and **5i**) derivatives of the synthesized series of disulfide conjugates showed the highest antimicrobial activity against the test strains of microorganisms (Table 2). In addition, bromine derivatives were found to be more selective for growth inhibition of Gram-positive bacteria and yeast (Figure 4). There was no clear effect of the carbohydrate fragment on the activity of the studied compounds, but among the most active, the xyloside derivative **5h** had the strongest growth inhibitory activity, with MIC_50_ of 1.15 (*S. aureus*), 1.56 (*C. albicans*), and 3.08 (*E. coli*) μM.

The synthesized disulfides were also tested for their ability to inhibit biofilm formation by test strains of microorganisms using the modified method of Walencka [39], and the data are presented in Table 2. It was shown that the studied series of synthesized disulfides (**4a–5j**) had less pronounced inhibition of biofilm formation compared to the inhibition of planktonic forms of microorganisms. This is most likely due to the fact that the compounds inhibit the growth of microorganisms and they do not have time to form a biofilm under the action of tested compounds. Bromine and chlorine derivatives of disulfides had the most pronounced inhibitory effect on biofilms of the test strains of microorganisms (Figure 4). The most pronounced inhibitory effect on biofilm formation of the yeast-like fungi *C. albicans* was demonstrated by bromine derivative **5h** and chlorine derivative **4i**, with MIC_50_ of 3.21 and 4.08 μM. Bromine derivatives **4h** and **5c** inhibited biofilm formation by *C. albicans* with MIC_50_ 7.11 and 8.69 μM, and chlorine derivative **4d** with MIC_50_ 7.03 μM. The studied series of synthetic disulfides had a more pronounced effect on the biofilm formation of the yeast-like fungi *C. albicans* in comparison with bacteria.

Among the studied disulfides, the most significant activity inhibiting the biofilm formation of Gram-negative bacteria *E. coli* was demonstrated by compounds **5j**, **5d**, **4c**, and **5h** with MIC_50_ of 5.62, 6.00, 8.04, and 8.92 μM, respectively (Table 2). The greatest activity in relation to the inhibition of biofilm formation by the Gram-positive bacterium *S. aureus* was shown by bromine derivatives **4h** and **4c**, with MIC_50_ of 6.32 and 7.94 μM and chlorine derivative **4i** with MIC_50_ of 6.25 μM.

## 3. Materials and Methods

### 3.1. General Information

All reagents and solvents were obtained from commercial suppliers and used without further purification. The starting bis(2-amino-4-phenyl-5-thiazolyl) disulfides **1a–e** [8,12], tetra-O-acetyl-1-thio-β-D-glucopyranose **3a**, and tri-O-acetyl-1-thio-β-D-xylopyranose **3b** [40] were obtained according to previously reported procedures. Melting points were determined using a Boetius apparatus and uncorrected. IR spectra in CHCl_3_ and KBr were obtained using a Bruker Vector-22 FT-IR spectrophotometer. ^1^H and ^13^C NMR spectra were recorded on Bruker Avance DPX-300 spectrometer. CDCl_3_ and DMSO-d_6_ were used as solvents, with SiMe_4_ (δ = 0 ppm) or the signal of the residual solvent (CHCl_3_: δ_H_ = 7.26, δ_c_ = 77.16, DMSO: δ_H_ = 2.50, δ_c_ = 39.52 ppm) as the internal reference. HRESIMS spectra were measured using an Agilent 6510 Q-TOF LC mass spectrometer. The reaction course was monitored by TLC. The copies of ^1^H NMR and ^13^C NMR spectra for synthesized compounds **4a–j**, **5a–j** can be found in Appendix A.

#### 3.1.1. General Procedure for the Synthesis of Acetylated Conjugates **4a–j**

Reduction of thiazole disulfides **1a–e**.

Sodium dithionite (85%, 8.2 g, 40 mmol, 20 eq) was added to a stirring suspension of the corresponding disulfide **1a–e** (2 mmol) in an acetone-water mixture (90 mL, 1:2 *v*/*v*) at 10 °C. The reaction mixture was stirred for 15 min at room temperature until a yellowish solution was formed. The mixture was diluted with cold water (100 mL) and cooled to 5 °C for 1 h. The precipitate of mercaptothiazole **2a–e** was filtered and washed with cold water (5 mL). The filtrate was extracted with chloroform (3 × 50 mL) and the extract was evaporated under reduced pressure at 40 °C. The yellowish residue containing compounds **2a–e** was combined with the filtered precipitate and immediately used in the next synthesis stage.

Oxidation of mercaptanes with I_2_.

To a mixture of acetylated mercaptosugar **3a** or **3b** (4 mmol) and corresponding freshly prepared mercaptothiazole **2a–e** in 70 mL of EtOH (mixture can be warmed to 40 °C for better solubility), a solution of I_2_ (0.8 M in 95% ethanol) was slowly added until a persistent yellow-brown color was observed. The reaction mixture was stirred for 15 min at room temperature, and the excess iodine was quenched with a sodium thiosulfate solution (5% in H_2_O) until a light yellow solution was formed. Sodium hydrogen carbonate (0.5 g) was added to neutralize the formed HI and the reaction mixture was stirred for 20 min. Inorganic salts were filtered off, the obtained solution was evaporated, and the residue was subjected to column chromatography (elution with a hexane-benzene-acetone mixture, 2:1:1 *v*/*v*) to obtain the corresponding conjugate **4a–j**.

5-[(2,3,4,6-Tetra-O-acetyl-β-D-glucopyranosyl)disulfanyl]-2-amino-4-(4-methoxyphenyl) thiazole (**4a**); yield 553 mg (23%, recovery of the starting disulfide **1a** 43%), white solid, mp 82–83 °C. ^1^H NMR (300 MHz, CDCl_3_) δ: 7.80 (d, *J* = 8.8 Hz, 2H, 2×Ar-H), 6.97 (d, *J* = 8.8 Hz, 2H, 2×Ar-H), 5.32 (s, 2H, -NH_2_), 5.04–5.16 (m, 2H, H^3^, H^4^), 5.01 (t, *J* = 9.4 Hz, 1H, H^2^), 4.48 (d, *J* = 9.4 Hz, 1H, H^1^), 4.26 (dd, *J* = 12.4, 4.8 Hz, 1H, H^6a^), 4.14 (dd, *J* = 12.4, 2.3 Hz, 1H, H^6b^), 3.86 (s, 3H, -OCH_3_), 3.49–3.54 (m, 1H, H^5^), 2.09 (s, 3H, -COCH_3_), 2.02 (s, 3H, -COCH_3_), 2.00 (s, 6H, 2×-COCH_3_). ^13^C NMR (75 MHz, CDCl_3_) δ: 171.1, 170.3, 169.4, 169.3 (×2), 160.1, 157.6, 131.1 (×2), 126.2, 113.7 (×2), 111.9, 87.4, 76.3, 73.9, 69.7, 68.3, 62.0, 55.4, 21.1, 20.7 (×3). IR (CHCl_3_, ν, cm^−1^): 3498, 3396, 3092, 3003, 2963, 2840, 1755, 1603, 1524, 1504, 1465, 1441, 1417, 1369, 1331, 1250, 1211, 1178, 1089, 1057, 1037. HRMS (ESI): *m/z* [M − H]^−^ calcd for C_24_H_27_N_2_O_10_S_3_: 599.0833; found: 599.0831.

5-[(2,3,4,6-Tetra-O-acetyl-β-D-glucopyranosyl)disulfanyl]-2-amino-4-phenylthiazole (**4b**); yield 456 mg (20%, recovery of the starting disulfide **1b** 45%), white solid, mp 114–115 °C. ^1^H NMR (300 MHz, CDCl_3_) δ: 7.78–7.82 (m, 2H, 2×Ar-H), 7.36–7.47 (m, 3H, 3×Ar-H), 5.36 (s, 2H, -NH_2_), 5.05–5.15 (m, 2H, H^3^, H^4^), 5.02 (t, *J* = 9.4 Hz, 1H, H^2^), 4.46 (d, *J* = 9.4 Hz, 1H, H^1^), 4.27 (dd, *J* = 12.4, 4.7 Hz, 1H, H^6a^), 4.12 (dd, *J* = 12.4, 2.1 Hz, 1H, H^6b^), 3.44–3.49 (m, 1H, H^5^), 2.10 (s, 3H, -COCH_3_), 2.03 (s, 3H, -COCH_3_), 2.00 (s, 6H, 2× -COCH_3_). ^13^C NMR (75 MHz, CDCl_3_) δ: 171.1, 170.3, 169.4, 169.3 (×2), 157.9, 133.8, 129.8 (×2), 128.9, 128.4 (×2), 113.6, 87.4, 76.3, 73.9, 69.7, 68.2, 62.0, 21.1, 20.7 (×3). IR (CHCl_3_, ν, cm^−1^): 3498, 3396, 3064, 3003, 1755, 1603, 1505, 1468, 1439, 1369, 1333, 1241, 1228, 1206, 1194, 1090, 1058. HRMS (ESI): *m/z* [M − H]^−^ calcd for C_23_H_25_N_2_O_9_S_3_: 569.0727; found: 569.0723.

5-[(2,3,4,6-Tetra-O-acetyl-β-D-glucopyranosyl)disulfanyl]-2-amino-4-(4-bromophenyl)thiazole (**4c**); yield 623 mg (24%, recovery of the starting disulfide **1c** 38%), white solid, mp 98–100 °C. ^1^H NMR (300 MHz, CDCl_3_) δ: 7.71 (d, *J* = 8.4 Hz, 2H, 2×Ar-H), 7.57 (d, *J* = 8.4 Hz, 2H, 2×Ar-H), 5.29 (s, 2H, -NH_2_), 5.08–5.20 (m, 2H, H^3^, H^4^), 5.04 (t, *J* = 9.4 Hz, 1H, H^2^), 4.50 (d, *J* = 9.4 Hz, 1H, H^1^), 4.31 (dd, *J* = 12.4, 4.9 Hz, 1H, H^6a^), 4.17 (dd, *J* = 12.4, 2.2 Hz, 1H, H^6b^), 3.55–3.60 (m, 1H, H^5^), 2.10 (s, 3H, -COCH_3_), 2.04 (s, 3H, -COCH_3_), 2.01 (s, 3H, -COCH_3_), 2.00 (s, 3H, -COCH_3_). ^13^C NMR (75 MHz, CDCl_3_) δ: 171.0, 170.3, 169.4, 169.3 (×2), 156.6, 132.8, 131.5 (×2), 131.3 (×2), 123.1, 114.2, 87.1, 76.5, 73.9, 69.7, 68.4, 62.1, 21.1, 20.7 (×3). IR (CHCl_3_, ν, cm^−1^): 3498, 3396, 3053, 2982, 2361, 1756, 1603, 1506, 1457, 1395, 1369, 1328, 1241, 1230, 1203, 1193, 1163, 1058, 1011. HRMS (ESI): *m/z* [M − H]^−^ calcd for C_23_H_24_BrN_2_O_9_S_3_: 646.9833; found: 646.9834.

5-[(2,3,4,6-Tetra-O-acetyl-β-D-glucopyranosyl)disulfanyl]-2-amino-4-(4-chlorophenyl)thiazole (**4d**); yield 460 mg (19%, recovery of the starting disulfide **1d** 47%), white solid, mp 116–118 °C. ^1^H NMR (300 MHz, CDCl_3_) δ: 7.76 (d, *J* = 8.6 Hz, 2H, 2×Ar-H), 7.41 (d, *J* = 8.6 Hz, 2H, 2×Ar-H), 5.39 (s, 2H, -NH_2_), 5.09–5.20 (m, 2H, H^3^, H^4^), 5.04 (t, *J* = 9.5 Hz, 1H, H^2^), 4.51 (d, *J* = 9.5 Hz, 1H, H^1^), 4.31 (dd, *J* = 12.5, 4.9 Hz, 1H, H^6a^), 4.17 (dd, *J* = 12.5, 2.2 Hz, 1H, H^6b^), 3.56–3.62 (m, 1H, H^5^), 2.10 (s, 3H, -COCH_3_), 2.04 (s, 3H, -COCH_3_), 2.01 (s, 6H, 2× -COCH_3_). ^13^C NMR (75 MHz, CDCl_3_) δ: 171.1, 170.3, 169.5, 169.4, 169.3, 156.6, 134.8, 132.2, 131.0 (×2), 128.6 (×2), 113.9, 86.9, 76.4, 73.8, 69.5, 68.3, 62.1, 21.1, 20.8 (×3). IR (CHCl_3_, ν, cm^−1^): 3498, 3396, 3020, 2959, 1753, 1603, 1506, 1459, 1432, 1400, 1369, 1328, 1247, 1193, 1164, 1092, 1058. HRMS (ESI): *m/z* [M − H]^−^ calcd for C_23_H_24_ClN_2_O_9_S_3_: 603.0338; found: 603.0333.

5-[(2,3,4,6-Tetra-O-acetyl-β-D-glucopyranosyl)disulfanyl]-2-amino-4-(p-tolyl)thiazole (**4e**); yield 608 mg (26%, recovery of the starting disulfide **1e** 37%), white solid, mp 90–92 °C. ^1^H NMR (300 MHz, CDCl_3_) δ: 7.67 (d, *J* = 8.1 Hz, 2H, 2×Ar-H), 7.25 (d, *J* = 8.4 Hz, 2H, 2×Ar-H), 5.74 (s, 2H, -NH_2_), 5.01–5.11 (m, 2H, H^3^, H^4^), 4.98 (t, *J* = 9.4 Hz, 1H, H^2^), 4.40 (d, *J* = 9.4 Hz, 1H, H^1^), 4.23 (dd, *J* = 12.4, 4.7 Hz, 1H, H^6a^), 4.10 (dd, *J* = 12.4, 2.0 Hz, 1H, H^6b^), 3.39–3.44 (m, 1H, H^5^), 2.39 (s, 3H, -CH_3_), 2.08 (s, 3H, -COCH_3_), 2.02 (s, 3H, -COCH_3_), 1.99 (s, 3H, -COCH_3_), 1.98 (s, 3H, -COCH_3_). ^13^C NMR (75 MHz, CDCl_3_) δ: 171.0, 170.3, 169.7, 169.4 (×2), 158.0, 138.9, 130.9, 129.6 (×2), 129.1 (×2), 112.5, 87.4, 76.2, 73.8, 69.6, 68.2, 62.0, 21.4, 21.0, 20.7 (×3). IR (CHCl_3_, ν, cm^−1^): 3499, 3396, 3061, 2959, 2361, 2342, 2361, 2342, 1755, 1603, 1504, 1469, 1369, 1329, 1306, 1242, 1230, 1214, 1193, 1090, 1057. HRMS (ESI): *m/z* [M − H]^−^ calcd for C_24_H_27_N_2_O_9_S_3_: 583.0884; found: 583.0879.

5-[(2,3,4-Tri-O-acetyl-β-D-xylopyranosyl)disulfanyl]-2-amino-4-(4-methoxyphenyl)thiazole (**4f**); yield 634 mg (30%, recovery of the starting disulfide **1a** 35%), white solid, mp 85–86 °C. ^1^H NMR (300 MHz, CDCl_3_) δ: 7.75 (d, *J* = 8.8 Hz, 2H, 2×Ar-H), 6.97 (d, *J* = 8.8 Hz, 2H, 2×Ar-H), 5.80 (s, 2H, -NH_2_), 5.06 (t, *J* = 8.5 Hz, 1H, H^3^), 4.97 (t, *J* = 8.6 Hz, 1H, H^2^), 4.68–4.76 (m, 1H, H^4^), 4.48 (d, *J* = 8.6 Hz, 1H, H^1^), 4.03 (dd, *J* = 11.6, 5.3 Hz, 1H, H^5a^), 3.85 (s, 3H, -OCH_3_), 3.13 (dd, *J* = 11.6, 9.6 Hz, 1H, H^5b^), 2.02 (s, 6H, 2×-COCH_3_), 2.00 (s, 3H, -COCH_3_). ^13^C NMR (75 MHz, CDCl_3_) δ: 170.1, 169.8, 169.5, 169.4, 160.1, 157.5, 131.0 (×2), 126.3, 113.7 (×2), 111.7, 87.9, 72.4, 69.5, 68.5, 65.9, 55.4, 20.8 (×3). IR (CHCl_3_, ν, cm^−1^): 3499, 3396, 2976, 2840, 1755, 1603, 1524, 1504, 1465, 1441, 1417, 1369, 1331, 1249, 1178, 1037. HRMS (ESI): *m/z* [M − H]^−^ calcd for C_21_H_23_N_2_O_8_S_3_: 527.0622; found: 527.0614.

5-[(2,3,4-Tri-O-acetyl-β-D-xylopyranosyl)disulfanyl]-2-amino-4-phenylthiazole (**4g**); yield 558 mg (28%, recovery of the starting disulfide **1b** 34%), white solid, mp 204–206 °C. ^1^H NMR (300 MHz, CDCl_3_) δ: 7.79–7.82 (m, 2H, 2×Ar-H), 7.38–7.48 (m, 3H, 3×Ar-H), 5.33 (s, 2H, -NH_2_), 5.05 (t, *J* = 8.5 Hz, 1H, H^3^), 4.97 (t, *J* = 8.5 Hz, 1H, H^2^), 4.67–4.74 (m, 1H, H^4^), 4.47 (d, *J* = 8.6 Hz, 1H, H^1^), 4.00 (dd, *J* = 11.6, 5.3 Hz, 1H, H^5a^), 3.09 (dd, *J* = 11.6, 9.6 Hz, 1H, H^5b^), 2.03 (s, 3H, -COCH_3_), 2.02 (s, 3H, -COCH_3_), 2.01 (s, 3H, -COCH_3_). ^13^C NMR (75 MHz, CDCl_3_) δ: 170.1, 169.8, 169.4, 169.2, 158.2, 133.9, 129.7 (×2), 128.9, 128.3 (×2), 113.8, 87.9, 72.4, 69.5, 68.5, 66.0, 20.9, 20.8 (×2). IR (CHCl_3_, ν, cm^−1^): 3501, 3398, 3106, 2985, 2869, 1754, 1603, 1507, 1460, 1371, 1328, 1245, 1193, 1164, 1122, 1092, 1069, 1035. HRMS (ESI): *m/z* [M − H]^−^ calcd for C_20_H_21_N_2_O_7_S_3_: 497.0516; found: 497.0512.

5-[(2,3,4-Tri-O-acetyl-β-D-xylopyranosyl)disulfanyl]-2-amino-4-(4-bromophenyl)thiazole (**4h**); yield 808 mg (31%, recovery of the starting disulfide **1c** 46%), white solid, mp 171–173 °C. ^1^H NMR (300 MHz, CDCl_3_) δ: 7.73 (d, *J* = 8.6 Hz, 2H, 2×Ar-H), 7.58 (d, *J* = 8.6 Hz, 2H, 2×Ar-H), 5.24 (s, 2H, -NH_2_), 5.09 (t, *J* = 8.5 Hz, 1H, H^3^), 5.00 (t, *J* = 8.6 Hz, 1H, H^2^), 4.66–4.74 (m, 1H, H^4^), 4.53 (d, *J* = 8.6 Hz, 1H, H^1^), 4.06 (dd, *J* = 11.6, 5.2 Hz, 1H, H^5a^), 3.15 (dd, *J* = 11.6, 9.4 Hz, 1H, H^5b^), 2.04 (s, 6H, 2× -COCH_3_), 2.01 (s, 3H, -COCH_3_). ^13^C NMR (75 MHz, CDCl_3_) δ: 170.1, 169.9, 169.4 (×2), 156.7, 132.8, 131.6 (×2), 131.2 (×2), 123.1, 114.1, 87.8, 72.2, 69.5, 68.5, 65.9, 20.8 (×3). IR (CHCl_3_, ν, cm^−1^): 3502, 3398, 3056, 2978, 2415, 2361, 1755, 1603, 1505, 1459, 1395, 1371, 1327, 1245, 1232, 1224, 1214, 1188, 1071, 1035, 1011. HRMS (ESI): *m/z* [M − H]^−^ calcd for C_20_H_20_BrN_2_O_7_S_3_: 574.9621; found: 574.9615.

5-[(2,3,4-Tri-O-acetyl-β-D-xylopyranosyl)disulfanyl]-2-amino-4-(4-chlorophenyl)thiazole (**4i**); yield 490 mg (23%, recovery of the starting disulfide **1d** 48%), white solid, mp 196–197 °C. ^1^H NMR (300 MHz, CDCl_3_) δ: 7.78 (d, *J* = 8.5 Hz, 2H, 2×Ar-H), 7.42 (d, *J* = 8.5 Hz, 2H, 2×Ar-H), 5.36 (s, 2H, -NH_2_), 5.10 (t, *J* = 8.5 Hz, 1H, H^3^), 5.01 (t, *J* = 8.5 Hz, 1H, H^2^), 4.66–4.74 (m, 1H, H^4^), 4.53 (d, *J* = 8.6 Hz, 1H, H^1^), 4.06 (dd, *J* = 11.6, 5.2 Hz, 1H, H^5a^), 3.16 (dd, *J* = 11.6, 9.4 Hz, 1H, H^5b^), 2.04 (s, 3H, -COCH_3_), 2.01 (s, 6H, 2× -COCH_3_). ^13^C NMR (75 MHz, CDCl_3_) δ: 170.1, 169.9, 169.4, 169.2, 156.7, 134.8, 132.4, 130.9 (×2), 128.6 (×2), 114.1, 87.7, 72.2, 69.4, 68.5, 65.9, 20.9 (×2), 20.8. IR (CHCl_3_, ν, cm^−1^): 3501, 3398, 3106, 2985, 2869, 1754, 1603, 1507, 1460, 1371, 1328, 1245, 1193, 1164, 1122, 1092, 1069, 1035. HRMS (ESI): *m/z* [M − H]^−^ calcd for C_20_H_20_ClN_2_O_7_S_3_: 531.0126; found: 531.0125.

5-[(2,3,4-Tri-O-acetyl-β-D-xylopyranosyl)disulfanyl]-2-amino-4-(p-tolyl)thiazole (**4j**); yield 543 mg (27%, recovery of the starting disulfide **1e** 44%), white solid, mp 199–200 °C. ^1^H NMR (300 MHz, CDCl_3_) δ: 7.69 (d, *J* = 8.1 Hz, 2H, 2×Ar-H), 7.26 (d, *J* = 8.1 Hz, 2H, 2×Ar-H), 5.50 (s, 2H, -NH_2_), 5.05 (t, *J* = 8.5 Hz, 1H, H^3^), 4.96 (t, *J* = 8.6 Hz, 1H, H^2^), 4.67–4.74 (m, 1H, H^4^), 4.45 (d, *J* = 8.6 Hz, 1H, H^1^), 4.01 (dd, *J* = 11.6, 5.3 Hz, 1H, H^5a^), 3.09 (dd, *J* = 11.6, 9.5 Hz, 1H, H^5b^), 2.39 (s, 3H, -CH_3_), 2.03 (s, 3H, -COCH_3_), 2.02 (s, 3H, -COCH_3_), 2.00 (s, 3H, -COCH_3_). ^13^C NMR (75 MHz, CDCl_3_) δ: 170.1, 169.8, 169.4, 169.3, 158.3, 138.9, 131.2, 129.6 (×2), 129.1 (×2), 112.9, 88.1, 72.5, 69.6, 68.6, 66.0, 21.5, 20.8 (×3). IR (CHCl_3_, ν, cm^−1^): 3501, 3397, 3114, 3019, 2983, 2869, 1755, 1602, 1505, 1470, 1432, 1371, 1329, 1306, 1248, 1193, 1118, 1068, 1035. HRMS (ESI): *m/z* [M − H]^−^ calcd for C_21_H_23_N_2_O_7_S_3_: 511.0673; found: 511.0669.

#### 3.1.2. General Procedure for the Synthesis of Deacetylated Derivatives 5**a–j**

The corresponding acetylated disulfide **4a–j** (0.5 mmol) was added to the solution of HCl in absolute MeOH (7 mL, ~1.6 M) and stirred for 15 h at room temperature. The solvent was removed under reduced pressure, and the residue was recrystallized from a mixture of MeOH-acetone (2:3 *v*/*v*, 5 mL) to give the hydrochloride of deacetylated product **5a–j**.

5-[(β-D-glucopyranosyl)disulfanyl]-2-amino-4-(4-methoxyphenyl)thiazole hydrochloride (5a); yield 211 mg (90%), white solid, mp 169–170 °C. ^1^H NMR (300 MHz, DMSO-d_6_) δ: 8.83 (br s, 1H, -N^+^H), 7.70 (d, J = 8.8 Hz, 2H, 2×Ar-H), 7.06 (d, J = 8.8 Hz, 2H, 2×Ar-H), 5.23 (br.s, 6H, 4× -OH, -NH_2_), 4.35 (d, J = 8.8 Hz, 1H, H^1^), 3.81 (s, 3H, -OCH_3_), 3.68 (dd, J = 11.8, 1.5 Hz, 1H, H^6a^), 3.52 (dd, J = 11.8, 5.0 Hz, 1H, H^6b^), 3.07–3.18 (m, 3H, H^2^, H^3^, H^4^), 3.00–3.04 (m, 1H, H^5^). ^13^C NMR (75 MHz, DMSO-d_6_) δ: 169.6, 160.1, 147.5, 130.6 (×2), 122.3, 113.9 (×2), 110.6, 88.5, 81.5, 77.9, 72.1, 69.5, 60.5, 55.4. IR (KBr, ν, cm^−1^): 3392, 3271, 3048, 2361, 2342, 1172, 1717, 1699, 1636, 1619, 1570, 1541, 1507, 1405, 1385, 1313, 1279, 1256, 1211, 1181, 1112, 1054, 1024, 883, 832, 736, 720. HRMS (ESI): m/z [M − H]^−^ calcd for C_16_H_20_ClN_2_O_6_S_3_: 467.0177; found: 467.0175.

5-[(β-D-glucopyranosyl)disulfanyl]-2-amino-4-phenylthiazole hydrochloride (**5b**); yield 202 mg (92%), white solid, mp 178–180 °C. ^1^H NMR (300 MHz, DMSO-d_6_) δ: 8.65 (br.s, 1H, -N^+^H), 7.72–7.75 (m, 2H, 2×Ar-H), 7.46–7.53 (m, 3H, 3×Ar-H), 5.95 (br s, 6H, 4× -OH, -NH_2_), 4.35 (d, *J* = 8.7 Hz, 1H, H^1^), 3.67 (dd, *J* = 11.8, 1.5 Hz, 1H, H^6a^), 3.50 (dd, *J* = 11.8, 5.0 Hz, 1H, H^6b^), 3.05–3.15 (m, 3H, H^2^, H^3^, H^4^), 2.99–3.03 (m, 1H, H^5^). ^13^C NMR (75 MHz, DMSO-d_6_) δ: 169.8, 148.2, 133.3, 129.5, 129.2 (×2), 128.4 (×2), 112.3, 88.6, 81.6, 77.9, 72.0, 69.5, 60.6. IR (KBr, ν, cm^−1^): 3375, 3275, 2932, 2892, 2360, 1628, 1587, 1570, 1543, 1488, 1448, 1425, 1385, 1323, 1278, 1207, 1183, 1091, 1061, 1016, 985, 919, 870, 788, 765, 732. HRMS (ESI): *m/z* [M − H]^−^ calcd for C_15_H_18_ClN_2_O_5_S_3_: 437.0072; found: 437.0070.

5-[(β-D-glucopyranosyl)disulfanyl]-2-amino-4-(4-bromophenyl)thiazole hydrochloride (**5c**); yield 243 mg (94%), white solid, mp 145–147 °C. ^1^H NMR (300 MHz, DMSO-d_6_) δ: 8.28 (br s, 1H, -N^+^H), 7.72 (d, *J* = 8.8 Hz, 2H, 2×Ar-H), 7.68 (d, *J* = 8.8 Hz, 2H, 2×Ar-H), 5.46 (br s, 6H, 4× -OH, -NH_2_), 4.33 (d, *J* = 8.7 Hz, 1H, H^1^), 3.65 (dd, *J* = 11.7, 1.5 Hz, 1H, H^6a^), 3.50 (dd, *J* = 11.7, 5.0 Hz, 1H, H^6b^), 3.07–3.26 (m, 3H, H^2^, H^3^, H^4^), 2.96–3.00 (m, 1H, H^5^). ^13^C NMR (75 MHz, DMSO-d_6_) δ: 169.8, 148.6, 131.3 (×2), 131.1 (×2), 130.9, 122.6, 112.5, 88.5, 81.6, 78.0, 72.1, 69.4, 60.5. IR (KBr, ν, cm^−1^): 3289, 2932, 2360, 2341, 1635, 1621, 1592, 1540, 1485, 1434, 1385, 1350, 1271, 1055, 1007, 915, 883, 824, 732, 712. HRMS (ESI): *m/z* [M − H]^−^ calcd for C_15_H_17_BrClN_2_O_5_S_3_: 514.9177; found: 514.9178.

5-[(β-D-glucopyranosyl)disulfanyl]-2-amino-4-(4-clorophenyl)thiazole hydrochloride (**5d**); yield 189 mg (80%), white solid, mp 149–150 °C. ^1^H NMR (300 MHz, DMSO-d_6_) δ: 8.40 (br s, 1H, -N^+^H), 7.80 (d, *J* = 8.5 Hz, 2H, 2×Ar-H), 7.55 (d, *J* = 8.5 Hz, 2H, 2×Ar-H), 5.47 (br s, 6H, 4× -OH, -NH_2_), 4.33 (d, *J* = 8.8 Hz, 1H, H^1^), 3.65 (dd, *J* = 11.8, 1.5 Hz, 1H, H^6a^), 3.50 (dd, *J* = 11.8, 5.0 Hz, 1H, H^6b^), 3.06–3.16 (m, 3H, H^2^, H^3^, H^4^), 2.95–3.00 (m, 1H, H^5^). ^13^C NMR (75 MHz, DMSO-d_6_) δ: 169.8, 148.9, 133.8, 130.9 (×2), 130.1, 128.4 (×2), 112.3, 88.5, 81.6, 78.0, 72.1, 69.4, 60.5. IR (KBr, ν, cm^−1^): 3372, 2362, 1637, 1621, 1560, 1385, 1349, 1271, 1095, 1054, 1016, 914, 882, 827, 732, 717. HRMS (ESI): *m/z* [M − H]^−^ calcd for C_15_H_17_Cl_2_N_2_O_5_S_3_: 470.9682; found: 470.9684.

5-[(β-D-glucopyranosyl)disulfanyl]-2-amino-4-(p-tolyl)thiazole hydrochloride (**5e**); yield 193 mg (85%), white solid, mp 175–176 °C. ^1^H NMR (300 MHz, DMSO-d_6_) δ: 8.84 (br s, 1H, -N^+^H), 7.62 (d, *J* = 8.0 Hz, 2H, 2×Ar-H), 7.32 (d, *J* = 8.0 Hz, 2H, 2×Ar-H), 5.82 (br s, 6H, 4× -OH, -NH_2_), 4.34 (d, *J* = 8.7 Hz, 1H, H^1^), 3.68 (dd, *J* = 11.8, 1.5 Hz, 1H, H^6a^), 3.51 (dd, *J* = 11.8, 5.0 Hz, 1H, H^6b^), 3.07–3.15 (m, 3H, H^2^, H^3^, H^4^), 2.99–3.04 (m, 1H, H^5^), 2.36 (s, 3H, -CH_3_). ^13^C NMR (75 MHz, DMSO-d_6_) δ: 169.7, 147.1, 139.5, 129.1 (×2), 129.0 (×2), 126.9, 111.9, 88.5, 81.6, 77.9, 72.0, 69.5, 60.5, 21.0. IR (KBr, ν, cm^−1^): 3277, 2907, 2361, 2343, 1717, 1699, 1628, 1586, 1566, 1542, 1504, 1435, 1385, 1346, 1320, 1273, 1209, 1186, 1115, 1086, 1057, 1021, 985, 870, 839, 817, 789, 734, 713. HRMS (ESI): *m/z* [M − H]^−^ calcd for C_16_H_20_ClN_2_O_5_S_3_: 451.0228; found: 451.0227.

5-[(β-D-xylopyranosyl)disulfanyl]-2-amino-4-(4-methoxyphenyl)thiazole hydrochloride (**5f**); yield 195 mg (89%), white solid, mp 149–150 °C. ^1^H NMR (300 MHz, DMSO-d_6_) δ: 8.73 (br s, 1H, -N^+^H), 7.70 (d, *J* = 8.8 Hz, 2H, 2×Ar-H), 7.06 (d, *J* = 8.8 Hz, 2H, 2×Ar-H), 5.72 (br s, 5H, 3× -OH, -NH_2_), 4.37 (d, *J* = 8.3 Hz, 1H, H^1^), 3.81 (s, 3H, -OCH_3_), 3.72 (dd, *J* = 11.1, 4.8 Hz, 1H, H^5a^), 3.18–3.26 (m, 1H, H^3^), 3.09–3.14 (m, 2H, H^2^, H^4^), 2.96 (t, *J* = 10.5 Hz, 1H, H^5b^). ^13^C NMR (75 MHz, DMSO-d_6_) δ: 169.5, 159.9, 149.1, 130.4 (×2), 123.2, 113.7 (×2), 110.1, 89.6, 77.0, 71.9, 69.0, 68.9, 55.3. IR (KBr, ν, cm^−1^): 3153, 3046, 2836, 2361, 2341, 1171, 1614, 1574, 1508, 1403, 1386, 1307, 1285, 1262, 1179, 1097, 1061, 1027, 968, 896, 831, 784, 740, 720. HRMS (ESI): *m/z* [M − H]^−^ calcd for C_15_H_18_ClN_2_O_5_S_3_: 437.0072; found: 437.0081.

5-[(β-D-xylopyranosyl)disulfanyl]-2-amino-4-phenylthiazole hydrochloride (**5g**); yield 192 mg (94%), white solid, mp 159–160 °C. ^1^H NMR (300 MHz, DMSO-d_6_) δ: 8.54 (br s, 1H, -N^+^H), 7.71–7.74 (m, 2H, 2×Ar-H), 7.46–7.53 (m, 3H, 3×Ar-H), 5.72 (br s, 5H, 3× -OH, -NH_2_), 4.35 (d, *J* = 8.3 Hz, 1H, H^1^), 3.70 (dd, *J* = 11.1, 4.8 Hz, 1H, H^5a^), 3.16–3.23 (m, 1H, H^3^), 3.07–3.13 (m, 2H, H^2^, H^4^), 2.93 (t, *J* = 10.8 Hz, 1H, H^5b^). ^13^C NMR (75 MHz, DMSO-d_6_) δ: 169.7, 148.7, 130.5, 129.5, 129.2 (×2), 128.4 (×2), 112.1, 89.6, 77.1, 71.9, 69.1 (×2). IR (KBr, ν, cm^−1^): 3371, 3279, 2926, 2859, 2684, 1670, 1625, 1592, 1572, 1542, 1487, 1458, 1421, 1385, 1343, 1280, 1240, 1214, 1182, 1158, 1095, 1061, 998, 965, 895, 818, 768, 731. HRMS (ESI): *m/z* [M − H]^−^ calcd for C_14_H_16_ClN_2_O_4_S_3_: 406.9966; found: 406.9968.

5-[(β-D-xylopyranosyl)disulfanyl]-2-amino-4-(4-bromophenyl)thiazole hydrochloride (**5h**); yield 222 mg (91%), white solid, mp 144–146 °C. ^1^H NMR (300 MHz, DMSO-d_6_) δ: 8.16 (br s, 1H, -N^+^H), 7.73 (d, *J* = 8.8 Hz, 2H, 2×Ar-H), 7.68 (d, *J* = 8.8 Hz, 2H, 2×Ar-H), 5.61 (br s, 5H, 3× -OH, -NH_2_), 4.35 (d, *J* = 8.3 Hz, 1H, H^1^), 3.71 (dd, *J* = 11.1, 5.0 Hz, 1H, H^5a^), 3.20–3.27 (m, 1H, H^3^), 3.08–3.16 (m, 2H, H^2^, H^4^), 2.93 (t, *J* = 10.3 Hz, 1H, H^5b^). ^13^C NMR (75 MHz, DMSO-d_6_) δ: 169.8, 150.0, 131.2 (×2), 131.1 (×2), 130.9, 122.4, 112.0, 89.6, 77.1, 72.0, 69.1 (×2). IR (KBr, ν, cm^−1^): 3367, 2361, 1626, 1566, 1542, 1508, 1487, 1458, 1435, 1385, 1292, 1211, 1184, 1088, 1053, 1007, 967, 897, 825, 712. HRMS (ESI): *m/z* [M − H]^−^ calcd for C_14_H_15_BrClN_2_O_4_S_3_: 484.9071; found: 484.9074.

5-[(β-D-xylopyranosyl)disulfanyl]-2-amino-4-(4-clorophenyl)thiazole hydrochloride (**5i**); yield 184 mg (83%), white solid, mp 135–137 °C. ^1^H NMR (300 MHz, DMSO-d_6_) δ: 8.07 (br s, 1H, -N^+^H), 7.78 (d, *J* = 8.5 Hz, 2H, 2×Ar-H), 7.55 (d, *J* = 8.5 Hz, 2H, 2×Ar-H), 5.96 (br s, 5H, 3× -OH, -NH_2_), 4.35 (d, *J* = 8.0 Hz, 1H, H^1^), 3.70 (dd, *J* = 11.2, 4.8 Hz, 1H, H^5a^), 3.18–3.25 (m, 1H, H^3^), 3.07–3.15 (m, 2H, H^2^, H^4^), 2.92 (t, *J* = 10.4 Hz, 1H, H^5b^). ^13^C NMR (75 MHz, DMSO-d_6_) δ: 169.8, 149.5, 133.8, 130.9 (×2), 130.4, 128.3 (×2), 112.1, 89.6, 77.1, 72.0, 69.1 (×2). IR (KBr, ν, cm^−1^): 3372, 1699, 1619, 1566, 1542, 1487, 1458, 1429, 1385, 1354, 1292, 1236, 1093, 1049, 1010, 964, 897, 826, 779, 733, 718. HRMS (ESI): *m/z* [M − H]^−^ calcd for C_14_H_15_Cl_2_N_2_O_4_S_3_: 440.9576; found: 440.9571.

5-[(β-D-xylopyranosyl)disulfanyl]-2-amino-4-(p-tolyl)thiazole hydrochloride (**5j**); yield 184 mg (87%), white solid, mp 134–136 °C. ^1^H NMR (300 MHz, DMSO-d_6_) δ: 8.64 (br s, 1H, -N^+^H), 7.62 (d, *J* = 8.0 Hz, 2H, 2×Ar-H), 7.31 (d, *J* = 8.0 Hz, 2H, 2×Ar-H), 5.53 (br s, 5H, 3× -OH, -NH_2_), 4.36 (d, *J* = 8.1 Hz, 1H, H^1^), 3.72 (dd, *J* = 11.1, 4.8 Hz, 1H, H^5a^), 3.18–3.26 (m, 1H, H^3^), 3.08–3.17 (m, 2H, H^2^, H^4^), 2.96 (t, *J* = 10.4 Hz, 1H, H^5b^), 2.36 (s, 3H, -CH_3_). ^13^C NMR (75 MHz, DMSO-d_6_) δ: 169.6, 148.1, 139.3, 129.0 (×2), 128.9 (×2), 127.3, 111.5, 89.6, 77.1, 71.9, 69.1 (×2), 21.0. IR (KBr, ν, cm^−1^): 3290, 2919, 2858, 1624, 1569, 1505, 1435, 1385, 1294, 1234, 1215, 1188, 1089, 1053, 1003, 957, 897, 818, 782, 733, 713. HRMS (ESI): *m/z* [M − H]^−^ calcd for C_15_H_18_ClN_2_O_4_S_3_: 421.0123; found: 421.0117.

### 3.2. Cell Lines

Human erythrocytes were purchased from the blood transfusion station Vladivostok. Human breast cancer cell lines MCF-7^HTB−22^, and MDA-MB-231^CRM-HTB−26^, liver hepatocellular carcinoma Hep G2^HB−8065^, prostate adenocarcinoma PC-3^CRL−1435^, cervical adenocarcinoma HeLa^CRM-CCL−2^, and normal bronchial epithelial cells BEAS-2B ^CRL−3588^ were received from ATCC (Manassas, VA, USA).

DMEM medium (Biolot, Russia) with 1% penicillin/streptomycin and 10% fetal bovine serum (FBS) (Biolot, Russia) was used for cultivation of Hep G2, PC-3, HeLa, and Beas-2b. MEM medium (Biolot, Russia) with 1% penicillin/streptomycin sulfate (Biolot, Russia) and FBS (Biolot, Russia) to a final concentration of 10% for MCF-7 and MDA-MB-231 cells.

### 3.3. MTT Assay to Determine the Effect on Cell Viability

The effect of the studied compounds on the viability of human cancer and normal cells was determined using the MTT (3-[4,5-dimethylthiazol-2-yl]-2,5 diphenyl tetrazolium bromide) assay. Briefly, 6 × 10^3^ cells per well were seeded into 96-well plates and incubated for 24 h for adhesion, followed by incubation for 24 h with different concentrations of the substances. After incubation, the medium containing the test substances was replaced with 100 µL of fresh medium. Then, 10 μL of MTT (Sigma-Aldrich, Madison, WI, USA) reagent (5 mg/mL MTT in PBS) was added into the cells and the plates for additional 4 h. Thereafter, 100 µL of SDS-HCl solution (1 g SDS/10 mL dH_2_O/17 µL 6N HCl) was added to each well and incubated for 18 h. The absorbance of the converted formazan dye was measured using a Multiskan FC microplate photometer (Thermo Scientific, Waltham, MA, USA) at a wavelength of 570 nm. All experiments were performed in triplicate. Cytotoxic activity was expressed as a percentage of the cell viability.

### 3.4. Wound Scratch Migration Assay

To analyze the effect of the tested compounds on the migration of HeLa tumor cells, special migration inserts (Culture-insert 2 Well 24, ibiTreat, Martinsried, Germany) were used, which left a gap of 500 ± 50 μm (according to the manufacturer’s data) between the cells attached to the plate. HeLa cells at a concentration of 5 × 10^5^ were added to each well of the migration insert for 24 h, the insert was removed, and the cells were washed twice with PBS to remove cell debris and floating cells and loaded with the fluorescent probe CFDA-SE (Lumiprobe, Moscow, Russia) at a concentration of 10 µM for 5 min. The cells were treated with different concentrations of the studied compounds for 0 and 24 h. Cells treated with culture medium alone were used as vehicle controls. Cell migration in the wound area was observed under a fluorescence microscope (MIB-2-FL, LOMO, St. Petersburg, Russia) with a 10× objective.

### 3.5. Colony Formation Assay to Determine the Effect on Cell Proliferation

The effect of the test compounds on HeLa cell proliferation was analyzed using a clonogenic assay. Briefly, HeLa cells were cultured in 6-well plates at a density of 1 × 10^2^ cells per well, and compounds were added at concentrations of 5 or 10 μM. Untreated culture medium was used as a control. The cells were incubated for 14 days at 37 °C with 5% CO_2_ until the cells in the control plates formed colonies that were visible to the eye and had a good size (at least 50 cells per colony). Before fixation and staining, the medium was removed and the cells were washed twice with PBS. The colonies were then fixed with methanol for 25 min and stained with 0.5% crystal violet for 25 min at room temperature. The plates were then washed with water and air-dried.

### 3.6. Antimicrobial Activity

The antimicrobial activity of the compounds was determined against Gram-positive bacteria *Staphylococcus aureus* (ATCC 21027), Gram-negative bacteria *Escherichia coli* (VKPM B-7935), and yeast-like fungi *Candida albicans* (KMM 455) (all test strains were obtained from the Collection of Marine Microorganisms KMM PIBOC FEB RAS) in liquid nutrient media. Test cultures of *S. aureus*, *E. coli*, and *C. albicans* were grown in Petri dishes on a solid nutrient medium (Mueller–Hinton broth with the addition of agar—16.0 g/L) at 37 °C for 24 h. The antimicrobial activity of the compounds was determined by the change in optical density and expressed as the percentage of bacterial inhibition at different concentrations of substances added to the well of the microorganism suspension, compared to the control [41]. The compounds were dissolved in dimethyl sulfoxide (DMSO) at a concentration of 10 mM and stored at −20 °C. Before the experiment, the solution of substances in DMSO was diluted with PBS at a concentration of 0.5–100.0 µM and added to the wells so that the concentration of DMSO in the culture medium was less than 1%. Antimicrobial activity was determined in 96-well microplates in a liquid Mueller–Hinton nutrient medium. For *S. aureus*, *E. coli*, and *C. albicans*, a bacterial suspension of 10^6^ CFU in 1 mL of the media was used, 90 μL was added to each well of the microplates and then incubated in a thermostatted shaker at 37 °C for 18 h. Inhibition of microorganism growth was assessed by measuring the optical density at 620 nm using a microplate reader (Thermo Fisher Scientific, Waltham, MA, USA) after preliminary shaking of the plates for 10 min. All experiments were performed in triplicate. Gentamicin (*S. aureus* and *E. coli*) and nitrofungin (*C. albicans*) were used as positive controls, and 1% DMSO in PBS was used as a negative control.

### 3.7. Inhibition of Biofilm Formation

Biofilm formation was performed according to the method described by Walencka et al. [39], with minor modifications. The compounds were dissolved in dimethyl sulfoxide (DMSO) at a concentration of 10 mM and stored at −20 °C. Before the experiment, the solution of substances in DMSO was diluted with PBS at a concentration of 0.5–100.0 µM and added to the wells so that the concentration of DMSO in the culture medium was less than 1%. Biofilm formation was assessed in 96-well microplates in a liquid Mueller–Hinton nutrient medium. For *S. aureus*, *E. coli*, and *C. albicans*, a bacterial suspension of 10^6^ CFU in 1 mL of medium was used, 90 μL was added to each well of the microplates and then incubated in a thermostatted shaker at 37 °C for 24 h. After incubation, the suspension of planktonic microorganisms was removed from the plate and washed with PBS. Next, 100 μL of PBS and 10 μL of MTT (3-(4,5-dimethylthiazol-2-yl)-2,5-diphenyltetrazolium bromide, MTT reagent, Sigma) stock solution (5 mg/mL) were added to the wells and incubated for 3 h at 37 °C. After incubation, all the solution was removed from the wells, 100 μL of DMSO was added, and the plate was shaken for 10 min until all formazan crystals had dissolved. The absorption was measured using a plate reader (Thermo Fisher Scientific, Waltham, MA, USA) at a wavelength of 570 nm. The activity of the substances was assessed by the change in optical density compared with that of the control. All experiments were performed in triplicate. Gentamicin (*S. aureus* and *E. coli*) and nitrofungin (*C. albicans*) were used as positive controls, and 1% DMSO in PBS was used as a negative control.

## 4. Conclusions

In this study, we developed a procedure for the synthesis of new carbohydrate-thiazole disulfides based on the alkaloid thiacarpine and its analogs **1a**–**e**. It was found that the type of sugar residue had a weaker influence on the activity of the synthesized compounds than the substituent type in the phenyl moiety of the thiazole ring. Acetylation of the sugar residue and the presence of –Me, –Br, and –OMe substituents of the aromatic part of the heterocyclic fragment are the most important factors for cytotoxic activity. The most promising compounds for the study of antitumor potential, **4c**, **4e**, **4f**, and **4h**, showed good migration inhibition and colony formation inhibition in cancer cell lines. The highest antimicrobial activity was observed for synthetic bromine **4c**, **4h**, **5c**, **5h**, and chlorine derivatives **4i** and **5d** which exhibited antimicrobial activity and inhibited biofilm formation by the Gram-positive bacteria *S. aureus*, Gram-negative bacteria *E. coli*, and yeast-like fungi *C. albicans*, comparable to the action of a standard antibiotic (gentamicin) and antimycotic (nitrofungin), but did not demonstrate hemolytic activity at concentrations up to 25 μM. The results also show that the introduction of carbohydrate fragments (glucose or xylose) into the molecule does not guarantee an increase in antimicrobial activity or selectivity of action on tumor cells, but in some cases, such a desired effect can be observed. Thus, of the studied substances, compound **5h** containing a xylose moiety, showed the most potent antimicrobial activity, on average twice as high as the starting symmetric disulfide **1c** and the comparison drugs.

## Data Availability

Data are contained within the article and Appendix A.

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
