# Peer review of "Synthesis and Biological Activity of Glycosyl Thiazolyl Disulfides Based on Thiacarpine, an Analogue of the Cytotoxic Alkaloid Polycarpine from the Ascidian *Polycarpa aurata"

_marinedrugs, 2025, doi:10.3390/md23030117_

Round 1

Reviewer 1 Report

Comments and Suggestions for Authors

Comments to Authors

The introduction should be supplemented with more information about thiacarpine and its biological activity.

L77,78 – Figure 1 should be moved as close as possible to the place of first mention.

Scheme 1 – yields of the synthesized compounds should be included.

Scheme 1:

  • condition i – ‘room temperature’ should be included.
  • condition iii – according to the Experimental, AcCl instead of HCl was used, and the time of the reaction was 15 h, instead of 16 h. Please check it and correct.

L110 – ….disulfide 1b ….

L110-114 - The experiments with bis(2-amino-4-phenyl-5-thiazolyl)disulfides 1b to test the typically effective alternative methods using sulfenyl halides 2b’, 3b’ showed that they were ineffective (yield of about 15-19% for paths B and C compared to 28% for path A) and more difficult to perform, as required anhydrous solvents and had lower solubility of the starting disulfide (Scheme 2) [37,38].

It probably needs to be rewritten and explained, because now it is unclear whether these experiments were presented in this text by the authors, or whether this is cited literature, because Ref. [37, 38].

Scheme 2 – in compound 4g, R2 should be removed.

L117, 118 – 4a-j and 5a-j instead of 4a-e and 5a-e. Next, according to Experimental section, AcCl in methanol instead of HCl in methanol was used.

L131 – Table 1 should be cited.

L174-177 – According to Table 2, bromo derivative 4c showed slightly higher activity against S. aureus than deacetylated 5c. Please include an explanation in the text of the manuscript.

Next, talking about series 4d, 4i, 5d, 5i, you should explain the reasons of the increase in activity for 5d>4d and decrease for 5i<4i.

In Table 2 – 4i – 8.71….

L217 – DMSO-d6 (6 in subscript).

In Experimental section – all multiplets should be characterized by intervals of values, but not by a single value. Next, all d.d should be changed with dd. Please check all 1H NMR spectra and correct it.

Section 3.1.2 – Were compounds 5a-j purified? Should be explained.

L355, 356 – please improve name of the compound 5a.

L493 – What solvent was used to dissolve compounds for antimicrobial testing? Have experiments been conducted to determine the inhibition of microbial growth and biofilm formation in this solvent? This should be included in the manuscript.

L534 – please correct the inaccuracy left.

Author Response

We thank the reviewer for the careful consideration of our manuscript. The reviewer's comments have been taken into account.

Comments 1: The introduction should be supplemented with more information about thiacarpine and its biological activity.

Response 1: We thank the reviewer for the careful consideration of our manuscript. The reviewer's comments have been taken into account. Some information about thiacarpine has been added.

Comments 2: L77,78 – Figure 1 should be moved as close as possible to the place of first mention.

Response 2: Figure 1 has been moved closer to the first paragraph

Comments 3: Scheme 1 – yields of the synthesized compounds should be included.

Response 3: The yields of the synthesized compounds have been included.

Comments 4: Scheme 1:

  • condition i – ‘room temperature’ should be included.
  • condition iii – according to the Experimental, AcCl instead of HCl was used, and the time of the reaction was 15 h, instead of 16 h. Please check it and correct.

Response 4: The required changes have been made. Acetyl chloride has been used as a convenient method for generating hydrogen chloride in methanol solution. The active reagent is hydrogen chloride at a concentration of about 1.6 M.

Comments 5: L110 – ….disulfide 1b ….

Response 5: Corrected.

Comments 6: L110-114 - The experiments with bis(2-amino-4-phenyl-5-thiazolyl)disulfides 1b to test the typically effective alternative methods using sulfenyl halides 2b’, 3b’ showed that they were ineffective (yield of about 15-19% for paths B and C compared to 28% for path A) and more difficult to perform, as required anhydrous solvents and had lower solubility of the starting disulfide (Scheme 2) [37,38].

It probably needs to be rewritten and explained, because now it is unclear whether these experiments were presented in this text by the authors, or whether this is cited literature, because Ref. [37, 38].

Response 6: This part has been rewritten.

Comments 7: Scheme 2 – in compound 4g, R2 should be removed.

Response 7: R2 has been removed in compound 4g.

Comments 8: L117, 118 – 4a-j and 5a-j instead of 4a-e and 5a-e. Next, according to Experimental section, AcCl in methanol instead of HCl in methanol was used.

Response 8: The required changes have been made. Experimental section has been modified.

Comments 9: L131 – Table 1 should be cited.

Response 9: Table 1 has been cited in this paragraph.

Comments 10: L174-177 – According to Table 2, bromo derivative 4c showed slightly higher activity against S. aureus than deacetylated 5c. Please include an explanation in the text of the manuscript.

Next, talking about series 4d, 4i, 5d, 5i, you should explain the reasons of the increase in activity for 5d>4d and decrease for 5i<4i.

Response 10: Indeed, for antimicrobial activity there is no general pattern in the change in activity between acetylated derivatives and deacetylated derivatives. Since the hybrid structures under study can be considered as multitarget, the resulting antimicrobial activity is the result of interactions with various targets in living cells. These interactions can cause effects that have different contributions and directions (increase or decrease toxicity). Accurate determination of the effect of substituents requires a detailed study of the mechanism of action of individual compounds.

Comments 11: In Table 2 – 4i – 8.71….

Response 11: Corrected.

Comments 12: L217 – DMSO-d6 (6 in subscript).

Response 12: Corrected.

Comments 13: In Experimental section – all multiplets should be characterized by intervals of values, but not by a single value. Next, all d.d should be changed with dd. Please check all 1H NMR spectra and correct it.

Response 13: Corrected.

Comments 14: Section 3.1.2 – Were compounds 5a-j purified? Should be explained.

Response 14: The compounds 5a-j purified by recrystallization. Experimental section has been modified.

Comments 15: L355, 356 – please improve name of the compound 5a.

Response 15: Corrected.

Comments 16: L493 – What solvent was used to dissolve compounds for antimicrobial testing? Have experiments been conducted to determine the inhibition of microbial growth and biofilm formation in this solvent? This should be included in the manuscript.

Response 16: The compounds were dissolved in dimethyl sulfoxide (DMSO) at a concentration of 10 mM and stored at -20 °C. Before the experiment, the solution of substances in DMSO was diluted with PBS at concentration 0.5-100.0 µM and added to the wells so that the concentration of DMSO in the culture medium was less than 1%.

Comments 17: L534 – please correct the inaccuracy left.

Response 17: Corrected.

Reviewer 2 Report

Comments and Suggestions for Authors

This manuscript describes an interesting approach to improving pharmaceutical properties of thiazolyl analogs of a tunicate natural product, polycarpine. Glycosylated derivatives possessing superior antimicrobial properties were synthesized and tested. However, the English narrative in the Abstract, Introduction , Results and Discussion sections needs to be improved (The Materials and Methods section is satisfactory). Tables 1 and 2 are difficult for the reader because of crowding of data. In Table 1 the use of "greater than 25" in most positions is distracting; can these be made to be less obtrusive? It is also suggested that three significant figures for an estimate should be sufficient, again making the data more easily digested by the reader. Similarly, the numbers in table 2 should be rounded off to three rather than four significant figures and this table should be enlarged too make it more intelligible.

Comments on the Quality of English Language

See the above section.

Author Response

We thank the reviewer for the careful consideration of our manuscript. The reviewer's comments have been taken into account.

Comments 1: However, the English narrative in the Abstract, Introduction, Results and Discussion sections needs to be improved (The Materials and Methods section is satisfactory).

Response 1: The article has been checked and corrected.

Comments 2: Tables 1 and 2 are difficult for the reader because of crowding of data. In Table 1 the use of "greater than 25" in most positions is distracting; can these be made to be less obtrusive? It is also suggested that three significant figures for an estimate should be sufficient, again making the data more easily digested by the reader. Similarly, the numbers in table 2 should be rounded off to three rather than four significant figures and this table should be enlarged too make it more intelligible.

Response 2: Tables 1 and 2 have been optimized. Duplicate values for rows and columns have been removed and appropriate footnotes added. The numbers in tables have been rounded off to three significant figures.

Round 2

Reviewer 1 Report

Comments and Suggestions for Authors

Thanks to the authors for correcting the article.

Reviewer 2 Report

Comments and Suggestions for Authors

The revised ms is now greatly improved and considered acceptable to this reviewer